# Evaluation of Tele-Dentistry and Face-to-Face Appointments during the Provision of Dental Services in Poland

**DOI:** 10.3390/jpm12101640

**Published:** 2022-10-03

**Authors:** Klaudia Migas, Remigiusz Kozłowski, Aleksandra Sierocka, Michał Marczak

**Affiliations:** 1Department of Management and Logistics in Healthcare, Medical University of Lodz, 90-419 Lodz, Poland; 2Center of Security Technologies in Logistics, Faculty of Management, University of Lodz, 90-237 Lodz, Poland

**Keywords:** telemedicine, COVID-19, remote consultation, dentistry

## Abstract

Tele-dentistry is a rapidly growing field, especially in the era of the COVID-19 pandemic. Due to the COVID-19 pandemic, remote services are of increasing interest to both patients and dental personnel. They allow for reduced person-to-person contact and thus a reduced risk of transmission of the SARS-CoV2 virus. The COVID-19 pandemic has affected the functioning of all areas of life, including dental treatment. The aim of the study was to assess the possibility of using tele-dentistry for dental services and analyse the attitudes of patients and dentists towards this solution. The period analysed was between March 2019 and February 2021 in five healthcare entities in Cracow in Central Europe. The study’s retrospective analysis shows a positive attitude of patients towards tele-dentistry at every stage of treatment, from diagnosis through postoperative care, and a significant reluctance of dentists in the majority of dental specialties towards tele-dentistry. Consequently, a significant percentage of patients were invited to dental offices for a face-to-face appointment during the COVID-19 pandemic. The negative attitude of dental personnel towards tele-dentistry compared with the positive attitude of patients towards tele-dentistry is somewhat worrying in view of the possibility of a further pandemic. At the same time, it provides important information about the need to educate and support dental personnel in tele-dental solutions and improve solutions for the future. Taking into account the potential reduction in dental care costs for patients and countries after the implementation of tele-dentistry solutions, this is an important topic, while current studies do not comprehensively address the attitudes of patients and dental personnel towards tele-dentistry. In other parts of the world, a similar approach to tele-dentistry is used by patients and dentists.

## 1. Introduction

The COVID-19 pandemic has had a significant impact on the functioning of virtually all areas of life. Its impact has changed the functioning of dental offices and the way dental services are provided [1,2,3].

According to the recommendations of the WHO, medical associations, including dental societies, and regulations at the state level, it was recommended to limit interpersonal contacts and to introduce the so-called remote, i.e., online, working.

Due to the significant development of dentistry in recent years and the implementation of digital solutions in dental treatment, both at the stage of diagnosis, therapy and postoperative care, it is possible to carry out certain dental procedures through tele-dentistry solutions [3,4].

Tele-dentistry enables a certain group of dental services to be carried out without direct contact with the patient. This advantage of tele-dentistry has enabled its more effective introduction in dental offices [5].

Tele-dentistry is used in various dental specialties, including conservative dentistry, endodontics, dental prosthetics, dental surgery, periodontics, orthodontics and implantology.

During the COVID-19 pandemic, entities performing dental services in Poland were advised to limit the number of patient dental appointments to the necessary minimum, especially for pain relief and dental assistance in order to reduce patient visits to the dental office. Recommendations on the operation of dental practices included not only limiting the number of appointments, extending the time between patient appointments, the nature of appointments, additional personal protective equipment and disinfecting dental offices, but also implementing remote patient admission. It is left to the dentist’s unlimited discretion to decide on the nature of the appointment, i.e., online consultation or face-to-face appointments. If it was not possible to help the patient through tele-dentistry methods, a visit to a dental office was recommended.

The sustainable development of tele-dentistry has been accelerated by the COVID-19 pandemic and the growth of digital services. The development of remote services may have a beneficial effect on many aspects of life, including easier access to medical and dental services [6,7]. In accordance with the FDI Consensus on Environmentally Sustainable Oral Heatlcare: A Joint Stakeholder Statement, March 2022, there is an opportunity to reduce the carbon footprint of dental offices through alternative modalities, such as tele-dentistry and remote dental appointments. Tele-dentistry can reduce face-to-face appointments and by this reduce amount of single use equipment and consequently wastes. Use of digital technologies in dentistry reduce the need for personal contact and unnecessary travel, which results in a smaller impact on air pollution. The adoption of digital technologies in dentistry needs to be carefully assessed because of the huge amount of equipment and technical support. Therefore, the aim of the study was to assess the possibility of using tele-dentistry for dental services and analyse the attitudes of patients and dentists towards this solution. In other parts of the world, a similar approach to tele-dentistry is used by patients and dentists [8,9].

In accordance with data from the Central Statistical Office in Poland, the number of working dentists in Cracow is 1584, which is two dentists per 1000 inhabitants; in Poland the number of dentists is 42,425, which is one dentist per 1000 inhabitants. When one compares the data with Eurostat for other European countries, in countries such as Norway and Germany, the number of dentists per 1000 inhabitants is eight.

In accordance with data of the Central Statistical Office, the number of dental offices in Poland is 4326 and in Cracow—224 dental offices, which accounts for 5.2% of Polish dental offices. Five dental offices which were qualified for the study provide a full range of dental services for every dental specialization, including 386 dentists, constituting 24% of Cracow dentists. In accordance with data of the Central Statistical Office in Poland, the number of inhabitants of Cracow in 2020 was 779,966 people, and the population of Poland in 2020 was 38,265,000 people. Cracow’s population accounted for 0.02% of the Polish population. In 2020, in accordance with data of the Central Statistical Office in Poland, the number of online consultations in dentistry amounted to 116,500 online consultations, which accounts for 0.4% of all online consultations provided in the field of the healthcare system in Poland. Malopolskie Voivodeship, whose voivodeship city is Cracow, was on the third position in the list of voivodeships where online consultations in the healthcare system was most often provided in Poland.

The aim of the study was to evaluate tele-dentistry in Poland and the approach of patients and dental staff. 

## 2. Materials and Methods

This work is a retrospective study which included data from medical records obtained from 1250 patients at 5 healthcare entities in Cracow in Poland in Central Europe. Data related to dentistry visits from the period of lockdown and one year before it, so the study applies to the period from March 2019 to February 2021. The lockdown periods in Poland lasted from March to April 2020, October to November 2020, December 2020 to January 2021 and March to April 2021. The lockdown periods were related to the introduction of state-wide and local restrictions by limiting movement, restricting the activities of particular services, and sanitary and hygiene orders. Because the study was based on the retrospective analysis of medical records, the Bioethics Committee’s consent was not required based on national legislation.

Data concerned the number of remote consultations (online consultations) initiated by patients, the number of face-to-face appointments which were the result of an online consultations and the lack of possibility of curing the patient’s ailment, the number of consultations during the pandemic period and pre-pandemic period, and the number of consultations following individual dental services. After every visit, both on-line and face-to-face, the patients were asked to fill an anonymous questionnaire on medical service quality. In addition, the dentists providing medical services were also asked to fulfil the questionnaire. The information from the medical service quality survey from the years 2019–2021 was also included into the data as additional information. Tele-dentistry services were provided by general dentists and specialists of dental surgery, dental prosthetics, endodontics and conservative dentistry; therefore, the analysed data relates only to those fields of dentistry. An online appointment was conducted via video call via mobile telephones. An appointment was conducted according to the patient’s needs. The doctor performed the analysis on the basis of medical records from previous appointments and the symptoms reported by the patient. An online appointment was always a second patient’s appointment.

The data were collected from the period from March 2019 to February 2021. The questionnaire was checked in January 2019 on a group of 124 patients regarding their satisfaction with dental online consultations. The ICC was measured for dentists assessing patients. The data were available from dental offices where patients were treated. All patients gave informed consent, and the data included in the study were in an anonymous form. The ICC was defined for observers who rated patient questionnaires that were collected from patients on a one-off basis to assess their satisfaction with the online appointments.

The questionnaire of medical service quality was designed for the purposes of the Medical Centre. The questionnaire was pre-tested on 124 patients before presenting it to the wider group of patients. The intraclass correlation coefficient (ICC) was used to assess the reproducibility of measurement. The satisfactory result was considered when the intraclass correlation coefficient (ICC) was >0.80. If a lower result was obtained, editing was performed.

[Fig jpm-12-01640-ch001] shows the possible actions of a dentist while providing the dental service and illustrates the procedure of the dentist after the performed dental treatment, when the patient reported to the dentist in the form of online consultation and the dental problem which the patient reported could not be solved by means of online consultation. [Fig jpm-12-01640-ch001] shows the situation when, after a face-to-face appointment to a dental office, the dentist is able to solve the patient’s dental problem which the patient reported. If the patient’s post-treatment problem could not be resolved at the online consultation, a face-to-face appointment was recommended by the dentist. The entire period of dental appointment time was 1 week to 2 weeks.

Data were collected in Microsoft Excel and then statistically processed in Statistica 12 (Statsoft) and Statxact (Cytel). The results were statistically significant if *p* < 0.05. Calculations were performed using Student’s *t*-test, the Shapiro–Wilk test and the Mann–Whitney test.

## 3. Results

### 3.1. Analysis of the Number of Consultations Provided in the Pre-Pandemic and Pandemic Periods

There was observed a statistically significant increase (*p* = 0.03) in the number of online consultations during the pandemic (P) compared to the pre-pandemic period (PP) (Figure 1). During P, a statistically significant decrease (*p* = 0.04) in the number of online consultations was observed only from December 2020 to January 2021 compared to the same period in PP (Figure 1). The greatest decrease in online consultations was recorded in November 2020 compared to November 2019 (Figure 1). In contrast, the greatest increase per month during P was recorded in September 2020 in comparison to September 2019 (Figure 1).

However, no statistically significant change was observed in the number of face-to-face appointments during P (Figure 2).

In addition, there was a no statistically significant increase (*p* = 0.06) in the necessity of the face-to-face follow-up appointments after a patient’s online consultation during P (Figure 3).

### 3.2. Analysis of the Type of Provided Consultations in Pre-Pandemic and the Pandemic Periods

The changes in the number of online consultations of different services were observed during P in comparison to PP. The changes were in the services as follows: dental surgery (extraction of a tooth or teeth) (Figure 4), dental prosthodontics (making a removable dental prosthesis (Figure 5) and a dental crown (Figure 6), microscopic endodontics (root canal treatment) (Figure 7) and conservative dentistry (making a light-cured filling) (Figure 8).

The list of the average of individual measured services for 2019 and 2020 as well as the *p*-value are shown in Table 1.

The highest increase in face-to-face follow-up appointments recommended by dentists after online consultation during P compared to PP was in May 2020. In contrast, during P, the highest decrease in face-to-face follow-up appointments occurred in August 2020 compared to August 2019.

There were changes in the reasons for online consultations after dental services provided during P compared to PP. For tooth extractions, there was the highest increase in online consultations in March 2020 and the highest decrease in July 2020 compared to PP. For dentures, the highest increase in online consultations was in July 2020 and the highest decrease in October 2020 compared to PP. Online consultations due to root canal treatment also changed during P compared to PP, and so the highest increase in online consultations was recorded in April 2020 and the highest decrease in November 2020. The highest increase in online consultations after performing a prosthetic tooth crown was recorded in July 2020 and the highest decrease in March 2020. In restorative dentistry, procedures such as the placement of a light-cured filling, changes in online consultations were recorded during P compared to PP with the highest increase in online consultations in September 2020 and the highest decrease in March and August 2020.

There was no reduction in the number of face-to-face appointments in the dental office, following the online consultation during P. All of the study patients in a situation where there was a possibility to provide dental care via online consultation were in favour of this form of appointment in comparison to dentists, of which only 61% were in favour of this form. There is a statistically significant correlation between the online consultations after a face-to-face appointments at a dental office; it is a strong and positive relationship; Pearson’s r correlation is 0.761, and the significance level is <0.001. There is a statistically significant correlation with a strong and positive relationship for variables in the form of online consultations with follow-up face-to-face appointments at a dental office; the Pearson’s r correlation is 0.786 and the significance level is <0.001.

## 4. Discussion

The COVID-19 pandemic has forced a change in functioning in almost every sphere of life, including the functioning of dental offices and patients’ perception of dental services [6,7,10,11].

Recommendations to maintain social distance and limited interpersonal contacts forced a change in the way of communicating and delivering services. Where face-to-face contact could be limited, remote communication was recommended. In the field of dentistry, not only was it recommended to function on the basis of the most necessary treatment, limiting appointments, appropriate personal protective equipment and extending the time between patient appointments for the recommended disinfection [12,13,14].

If it is possible to replace a face-to-face appointment with an online consultation, this solution was recommended. It is always the dentist who decides on the form of the appointment in order to achieve the optimal therapeutic effect. Online consultations in dentistry form an important part of the functioning of dental practices [15,16,17].

The study is unique in the sense that current publications cover the topic of telehealth, including tele-dentistry, and include the topic of COVID-19. However, this study assesses the change in the number of online consultations and the need for face-to-face appointments during the COVID-19 pandemic period in comparison with the pre-pandemic period. Importantly, the study illustrates how often in the case of online consultations in the post-pandemic period there is a need for face-to-face appointments, which is not found in other publications.

Dental personnel are divided over the effectiveness of online consultations and the provision of health services in this manner, including postoperative care. Taking into account the risk of a future pandemic, this should be regarded as worrying. Only 61% of the dentists who participated in the study supported a form of online consultation as a dental appointment and at the same time, during the COVID-19 pandemic period, there was no statistically significant increase in face-to-face follow-up appointment after an online consultation. It can be stated that if this increase was statistically significant in the months of May and June 2020, this is the initial period of the pandemic in which dentists were starting to function in such a capacity, despite the fact that prior to the COVID- 19 pandemic they were also providing online consultation services [6,18,19,20].

It is important to note that despite the lack of willingness of the staff to make online appointments, their statistically significant increase took place. The effectiveness of these appointments should be emphasized due to the lack of a statistically significant increase in follow-up face-to-face appointments.

Furthermore, other studies show that patients prefer to undergo online consultations, if possible [21].

An important fact is that the patients participating in the study prefer and support online consultations, preserving the quality of treatment if possible. Online consultations can be fully satisfactory for patients [22,23,24].

The sustainable development of tele-dentistry, which has now been accelerated, should be viewed positively. The development of digital services may have beneficial effects on many aspects of life, including easier access to medical services, including dental services, with great benefits for patients and medical staff.

Dentists do not feel confident in diagnosis without the possibility of direct examination of the patient. Merely relying on a medical history examination without the possibility of performing a physical examination may lead to a lack of complete diagnosis and given the responsibility for treatment, it may cause anxiety among dentists. The inability to perform a physical examination eliminates important results of the examination performed by dentists from the diagnosis, which significantly hinders the correct diagnosis and thus the implementation of effective treatment. In a situation where there is a high level of approval from patients but a lack of full approval from dental personnel, the personnel should be educated on the scope of remote devices, their effectiveness as well as solutions and safety of the provision of dental services. The active participation and involvement of dental personnel with an approval of up to 61% is a good start for developing tele-dentistry services. Tele-dentistry is a modern tool that is not based on the good evidence- based medicine, so it could be a huge disadvantage to use it during the diagnostic and treatment process and may not be accepted by dental staff [25,26,27,28].

It should be stated that it would be meaningful to carry out a study on patients and dentists in other countries for a full comparison and a post-pandemic study to assess the maintenance of the effectiveness of online consultations and the preferences of patients and dental personnel for this type of service. Tele-dentistry can be used in follow-up appointments after dental procedures, including surgery, and can be an alternative to face-to-face orthodontic follow-up appointments, especially during treatment with appliances [29,30].

The reluctance of dentists to conduct tele-dentistry appointments may be due to a lack of adequate preparation, training and education to effectively conduct such appointments. Other studies also confirm the lack of full willingness of dental personnel regarding remote tools, including online consultation [16,31,32,33].

During the pandemic, digital solutions such as tele-dentistry should be considered a helpful tool [34,35,36].

Technological advances are fostering the development of new remote working tools, including digital solutions, which can make it easier for the dentist to diagnose a problem in the patient’s mouth and thus reduce the need for the patient to attend a face-to-face appointment. The emergence of new digital solutions not only involving conversation with the patient but also the possibility of digital transmission of photographs, films or scans by the patient is expanding the possibilities of online consultation and increasing the information obtained by dental personnel from patients. Further development of digital tools that facilitate remote imaging of the patient’s condition increases the amount of information received by the dentist and thus may result in a reduction in the reluctance of dental personnel towards online consultation. The benefits of online consultation are not only a reduction in interpersonal contact during the pandemic, but at the same time a reduction in the cost of patient appointments, including through lower labour costs of the dental office and thus a lower cost per potential appointment and a reduction in the cost to the patient of travelling to the dental office. Tele-dentistry in the future will become more and more popular and an effective tool in the diagnostic and a treatment process [37,38,39].

A reduction in the reluctance of dental personnel to implement digital solutions and an increase in the effectiveness of online appointments can occur due to the development of digital technology and the introduction of new possibilities for conducting online consultations, e.g., a patient during a consultation can conduct a live video transmission or take a picture with his/her phone and send it to the dental personnel, who by having access to more information, including digital information, can decide on the effectiveness of an online appointment without having to make recommendations for a face-to-face appointment [40,41].

The patient can use his/her own home devices, such as a smartphone or laptop, for an online consultation. The development of these devices is not limited by tele-dentistry, which involves the creation of more and more applications that can enable remote communication. On the other hand, there are medical applications for dental personnel that allow processing data and receiving it in digital form, thus enabling a more accurate diagnosis and treatment of the patient [42,43,44,45].

The development of tele-dentistry and digital solutions in dentistry and medicine has been going on for a long time and it is anticipated that current technological progress will play an increasingly important role in modern dentistry and other medical fields [46,47,48,49,50,51].

Facing new challenges not only in dentistry, pandemics, but also in the environment, tele-dentistry is an opportunity for significant support and sustainable development on environmental issues. Reducing the consumption of dental materials and disposable materials used during stationary visits and the possibility of reducing pollution caused by reduced travel and exhaust fumes on the way to and from the dentist’s office contribute to a beneficial and sustainable impact on the environment.

The strength of this study is illustrating the perception of tele-dentistry appointments by patients and dental staff for various dental specialties, which is not found in other studies. At the same time, a comparison of these behaviors, i.e., increased recommendations for tele-dentistry appointments, between the pandemic period and the pre-pandemic period allows assessing the behavior of patients and staff in a specific and tough period.

However, some limitations of this study have to be emphasized, including the lack of follow-up in the post-pandemic period, which is a challenge for the future to conduct further studies on the perception of tele-dentistry by patients and dental staff, and the sample size of the study.

## 5. Conclusions

It is important to implement education and support dental personnel in the provision of remote services, the so-called online consultations. Along with the positive feedback of patients, they forms an important prognosis for the future and a good foundation for successful online consultations. The current reluctance of dental personnel towards online consultations is unfavourable, both in terms of safety of patients in the event of further pandemics and also for financial reasons regarding patients.

## Data Availability

Non-digital data supporting this study were curated by Klaudia Migas.

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
