# Peer review of "Evaluation of Tele-Dentistry and Face-to-Face Appointments during the Provision of Dental Services in Poland"

_jpm, 2022, doi:10.3390/jpm12101640_

Round 1

Reviewer 1 Report

Attached is the PDF file with revisions to be made.

Reviewer 2 Report

Dear Authors, 

Teledentistry has emerged as important tool during Pandemic. The manuscript has addressed the teledentistry option and its acceptance/ reluctance during pandemic . In my opinion few of the concerns which must be addressed are as follows:

Title : This article addresses the situation in Poland , I would like to suggest the authors to rephrase the title and be more specific for writing Poland . The current title looks like a review article.

Introduction:

1.       Kindly mention the study/studies which followed a similar study protocol ( Tele dentistry in another part of the world ). How teledentistry is perceived in other parts of the world ?

Material and methods :

Both diagrams are confusing. The labels are not self-explanatory. Kindly elaborate .

Results : p-value should be in points. kindly replace the commas(,)

Discussion:

The authors should also discuss the drawbacks of tele dentistry. As it can be one reason for non-acceptance.

The authors need to discuss or provide data for undiagnosed cases of lesions or dental problems during an online consultation.

Regards  

Reviewer 3 Report

Dear authors,

This study is a timely theme at a time when the demand for telemedicine is growing. The authors believe that tele-dentistry-related research for dental services using patient record data is meaningful, but the following checks are necessary for the contents to be equipped as a research article.

 1. Abstract

- The contents of the research method are insufficient. The description of the research tools is particularly lacking. Considering the limited size of the abstract, the research background should be more concisely summarized and the research method should be further described.

 2. Instructions to authors need to be reviewed for compliance

- In the title, the first letter of all words except articles and conjunctions should be capitalized.

 - Observe the regulations for reference numbering. ex) [1] [2] [3] to [1-3]; [3] [4] to [3,4]

 - Write all keywords in lower case except for proper nouns.

 3. Introduction

- The purpose of the study is unclear. Please clearly state the research purpose for the research results derived from this study.

 4. Methods

- It is ambiguous to understand the flow of research only with Diagrams 1 and 2. Please describe more clearly, including the number of study subjects in the research procedure.

- When the description of the variables measured in this study is specifically described, the validity of the research results can be secured.

- Even if IRB approval is exempted for not using personal data, please add content for ethical consideration of research subjects.

 5. Results

- In <Table 1>, check the punctuation points and present the mean with the standard deviation. ex) 793,1 to 793.1

- Also check the punctuation for the p-value notation. 0,01 to 0.01

- Describe the statistical analysis method at the bottom of the table.

 6. Discussion

Please supplement the limitations of this study and suggest future research based on this.

Best regards,

Reviewer 4 Report

I carefully reviewed this paper. It just piled up a few figures on the study of Tele-density before and during the Covid pandemic. I think this topic is very interesting, but the results of this paper are not conclusive. The author should provide more data to show why did happen. If possible, the author should send questionnaires to the patients and get their feedback. For example, the author can design the questionnaires why the patients choose online consultation but not face to face treatment, etc. What’s the key factors affect the patient’s decision, etc. However, I didn’t see any supporting data like this except the summary of service number month by month. To enrich the content of this paper, I suggest the author provide more meaningful data rather than providing redundant figures. Besides this, two comments on making this paper clear are listed below:

1.       The diagram 1 and 2 are very confusing for the reviewer to understand. Please provide more details or make the diagram clear for the reviewer.

2.       In the bar curves, I suggest the author find a way to highlight the data during the lockdown so that it’s easier for the readers to read those data.

Round 2

Reviewer 3 Report

Dear authors,

Overall, the manuscript has revised well.

But please describe the statistical analysis method in the table footer.

And table title must be at the top of the table. 

Best regard,

Reviewer 4 Report

Except for minor language problems, this paper is ready to be published.
